# Effectiveness of the Muscle Energy Technique versus Osteopathic Manipulation in the Treatment of Sacroiliac Joint Dysfunction in Athletes

**DOI:** 10.3390/ijerph17124490

**Published:** 2020-06-22

**Authors:** Urko José García-Peñalver, María Victoria Palop-Montoro, David Manzano-Sánchez

**Affiliations:** 1Facultad de Fisioterapia, Universidad Católica de San Antonio (UCAM), Av. de los Jerónimos, 135, 30107 Murcia, Spain; urkojgarcia@gmail.com (U.J.G.-P.); mvpalop@ucam.edu (M.V.P.-M.); 2Facultad de Ciencias del Deporte, Universidad de Murcia, Calle Argentina, 19, 30720 San Javier, Murcia Spain

**Keywords:** sacroiliac joint dysfunction, joint mobilization, thrust, muscle energy technique, athletics

## Abstract

Background: The study of injuries stemming from sacroiliac dysfunction in athletes has been discussed in many papers. However, the treatment of this issue through thrust and muscle-energy techniques has hardly been researched. The objective of our research is to compare the effectiveness of thrust technique to that of energy muscle techniques in the resolution of sacroiliac joint blockage or dysfunction in middle-distance running athletes. Methods: A quasi-experimental design with three measures in time (pre-intervention, intervention 1, final intervention after one month from the first intervention) was made. The sample consisted of 60 adult athletes from an Athletic club, who were dealing with sacroiliac joint dysfunction. The sample was randomly divided into three groups of 20 participants (43 men and 17 women). One intervention group was treated with the thrust technique, another intervention group was treated with the muscle–energy technique, and the control group received treatment by means of a simulated technique. A prior assessment of the range of motion was performed by means of a seated forward flexion test, a standing forward flexion test, and the Gillet test. After observing the dysfunction, the corresponding technique was performed on each intervention group. The control group underwent a simulated technique. A second intervention took place a month later, in order to ascertain possible increased effectiveness. Results: Statistically significant differences were found between the muscle energy technique (MET) and muscle energy groups compared with the placebo group in both interventions (*p* = 0.000), with a significant reduction in positive dysfunction (initially 20 in all groups, eight in MET group, and two in thrust group in the final intervention). Comparing the changes in time, only the thrust group obtained statistically significant differences (*p* = 0.000, with a reduction of positive dysfunction, starting at 20 positives, five positive in the initial intervention and two positive in the final intervention) and when comparing both techniques, it was observed that between the first intervention and the final intervention, the thrust technique was significantly higher than the MET technique (*p* = 0.032). Conclusions: The thrust manipulation technique is more effective in the treatment of sacroiliac dysfunction than the energy muscle technique, in both cases obtaining satisfactory results with far middle-distance running athletes. Finally, the thrust technique showed positive results in the first intervention and also in the long term, in contrast to the MET technique that only obtained changes after the first intervention.

## 1. Introduction

The sacroiliac joint (SIJ) is a region that includes multiple bone, joint, cartilage, muscle, ligament, and nerve structures. Its anatomical integrity and proper interaction enable the individual to perform normal daily activities [1].

SIJ is designed to provide stability and plays a key role in the development of pelvic joint biomechanics [1]. One of its main functions is to transfer weight from the trunk to the lower limbs. Thus, a SIJ dysfunction can lead to an imbalance in the mobility of lower limbs, interfering with the way someone walks and runs [2].

There are numerous intra and extra-articular etiologies leading to a SIJ dysfunction [3]. Risk factors include lumbar sacral spinal fusion, different length of the lower limbs, scoliosis, gait abnormality, and prolonged vigorous exercise [3].

Pain due to this dysfunction can be located at or around the lower back, hip, leg, foot, and toes [4]. Thus, the prevalence of lower back pain due to this sacroiliac dysfunction is at 22.5% in the adult population [5]. The prevalence of chronic lower back pain due to SIJ is 13%–30% [6]. In some cases, however, the dysfunction may be asymptomatic, with the individual not reporting any pain or discomfort [5].

Despite the high prevalence of sacroiliac dysfunction, there are no therapeutic protocols for this syndrome. It has been previously considered as a potential contributor to lower back pain but, according to scholars, there are few studies that have compared the effectiveness of the different therapeutic techniques [7].

Sacroiliac joint dysfunction is a common cause of pain amongst athletes. It is therefore important to consider this population group, since sport can sometimes predispose athletes to suffer from this type of pathology [8]. In particular, SIJ dysfunction in runners may be due to the impact of the lower limbs on the ground, which transfers in turn to the joint, eventually leading to a blockage that interferes with the correct biomechanics [9].

Diagnosis of the etiology is problematic because there are no pathognomonic signs, so a combination of clinical tests is needed in order to make a more accurate diagnosis. It has been proven that the performance of three or more positive maneuvers reports a sensitivity of 94% and a specificity of 78% [10].

The muscle energy technique (MET) and the thrust manipulation technique are among the most commonly mentioned in several studies on the treatment of sacroiliac dysfunction. The MET technique helps correct the dysfunction by means of isometric and isotonic contractions resulting in an improved physiological function of the joint [1,11].

The manipulation technique, or thrust, consists of a high-speed movement aimed at correcting the dysfunction in the direction of correction, which leads to both, joint improvement and periarticular muscle inhibition [3,12]. According to the authors, manipulative techniques have excellent results after short-term manipulation but their long-term benefits and recurrence prevention effects remain unproven [13]. Some authors claim in their reviews that the manipulation technique is the most efficient in the treatment of sacroiliac dysfunction [14].

In the light thereof, we can state that SIJ is an important anatomical element in the stride progression, being key to the correct distribution of body mass in both limbs by maintaining the stability and alignment of the lower limb. The limitation of its mobility can lead to an alteration in the correct execution of the gait and the stride [15]. Therefore, the objective of this study was to compare the effectiveness of thrust and MET techniques in the resolution of SIJ dysfunction in middle-distance running athletes.

## 2. Materials and Methods

### 2.1. Design Research

A quasi-experimental study was carried out. A measurement of sacroiliac dysfunction was performed before the first intervention (pre-intervention), a second measurement after the first intervention (post-intervention 1) and a last measurement one month after the first intervention (post-intervention 2). Athletes filled out a questionnaire aimed at finding out about a series of sociodemographic variables (Table 1 and Table 2). The existence of sacroiliac blockage, as well as which limb was suffering from it, were analyzed as independent variables. A random selection of participants was made in three groups. Each individual was entered into an Excel database and, by using the random number generation system, they were assigned a number (from 1 up to 60) prior to the manipulation. Then, three groups of 20 participants were made.

### 2.2. Sample

The sample was made up of 60 individuals, 43 male and 17 female, aged 18–62 (average age being 33.86 years, SD = 9.89), with an average body mass of 71.52 kg (SD = 12.67) and height of 1.75 m (SD = 8.78). Six were left-handed and 54 were right-handed individuals. All of them had limited mobility as evidenced by the exploratory tests. Blockage or dysfunction was initially observed on the left side in 46 individuals, while a right-side dysfunction was observed in 14 participants.

In Table 1 and Table 2, we can see the sociodemographic variables of the sample. Namely, the quantitative (Table 1) and categorical (Table 2) variables. In Table 1, average and standard deviation for the variables in each intervention group are shown, namely age, body mass, height, weekly training frequency, and frequency of visits to the physical therapist in days per year. Table 2 presents the results percentages and the total count of the categorical variables. The chi-square test shows the differences between the sexes (more males than females), work situation (working people were in the majority), dominant leg (most were right dominant leg), use of insoles (most of them do not use insoles), and previous injures (most of them had no injury record), versus those who had injuries in left leg, right leg or in both.

The inclusion criteria were athletics athletes who ran middle distance (minimum of 30 km per week), aged 18–63, and presenting a dysfunction in the SIJ of iliac origin.

The exclusion criteria were lower back pain, vertebral bone pathology, radiculopathy, recent fracture or surgery in the lumbosacral or pelvic area, anatomical leg length inequality with a difference greater than 0.5 cm, pregnancy, infection, and fear of the maneuver [12].

The deontological set of ethical principles recognized by the Declaration of Helsinki (64th General Assembly, Fortaleza, Brazil, October 2013) were respected. Each participant was informed of the objective of the study and received information on its development, specifically on the SIJ, the procedure of each of the techniques used, and the measurements to be taken. Any possible doubts were clarified at all times. In addition, written informed consent was provided to those who wished to participate in the study. The project was approved by the Ethics Committee of the Catholic University of Murcia (date: 1 March 2019; Code: CE031911).

### 2.3. Material and Procedures

The independent variables are: age, height, body mass, training frequency, frequency of visits to the physiotherapist of the participants in each intervention group, work situation, postural system normally used in the work place, dominant leg, previous injuries, use of insoles, and sports frequently practiced. They also signed an informed consent form.

Then, they were individually classified into groups and the corresponding intervention protocol was followed. Only one technique per intervention was performed in each group, so as to compare the techniques exclusively, without any other variable. These techniques were performed in a conditioned room of the athletic track, using a common clinical practice stretcher.

#### 2.3.1. Diagnosis

The following three validated tests were used for the analysis of mobility in SIJ: the Gillet test, the standing forward flexion test, and the seated forward flexion test. All three were chosen because they are commonly used for the evaluation of SIJ.

The Gillet test is performed by placing the patient in a standing position with the legs about 12 inches apart. The therapist palpates the S2 spinous process and the posterior superior iliac spine (PSIS). The patient is asked to raise his or her leg as if he or she was taking a big step. The test is considered positive if PSIS is not displaced caudally with respect to the sacrum [16,17,18].

The standing forward flexion test is performed with the patient standing. The therapist places his or her hands on the patient’s PSIS and the patient is asked to flex the trunk as far as possible. The result is positive if one of the PSISs moves more cranially than the contralateral [16,17,18]. The seated forward flexion test is performed in the same way but in this case the patient is seated.

#### 2.3.2. Intervention

The intervention followed the chronological temporalization of Figure 1. The MET technique for correction of anterior dysfunction of the sacroiliac joint was performed by placing the individual in a lateral position on the opposite side to that of the dysfunction. The blocked leg was taken by the examiner, while the unaffected leg remained extended on the stretcher. The leg was placed in hip flexion until the first point of tension that prevented the posterior rotation of the ilium was found. While this technique was being used, the patient was asked to push his leg into hip extension while the examiner held it and prevented movement. Four contractions were performed, resisted by the therapist, and held for 7–10 s. The individual was then asked to relax the leg and the examiner then continued to perform the hip flexion until a new point of tension was found. This was aimed at trying to get the ilium to rotate posteriorly. It was performed on three occasions, so that the anteriorly rotated ilium was pushed back in its position, thus releasing the blockage causing the anterior dysfunction of the sacroiliac joint [19].

In order to correct the SIJ dysfunction due to a posteriorly rotated ilium, the individual was placed in a prone position with his or her unaffected side leg in extension, while the affected side leg was in knee flexion. The therapist held the leg in a 90° flexion and tried to help hip extension so as to get the ilium to rotate posteriorly in order to correct the dysfunction. As soon as the examiner reached the first point of tension, he stopped and asked the individual to contract in a resisted way into the direction of hip flexion (four contractions during 7–10 s). After the contraction, the therapist advanced in the direction of the hip extension, looking for the anterior rotation of the ilium and a new point of tension. Once the second point of tension was found, the above process was performed, up to a total of three times [19].

The placebo technique was performed by placing the individual in a lateral position on the affected leg, performing a hip flexion and a 90° knee flexion sustained for 20 s, so that the contralateral leg was not affected and the mobility of the contralateral SIJ was not affected.

### 2.4. Reliability Control

The therapist who performed the tests and techniques was trained in the Master’s Degree in Osteopathy at the Catholic University of Murcia, where he followed a training process to put the tests into practice in accordance with the procedures recommended by different authors [16,17,18,20] He also performed the tests following the procedure which had been successfully applied in previous studies [19,21,22]. 

In order to ensure reliability and training of the therapist, two reliability criteria were followed:

Inter-observer reliability was tested by evaluating his ability to perform the thrust and the MET techniques. He was tested for these techniques, as well as for the Gillet test, the standing forward flexion test, and the seated forward flexion test, along with 5 other peers. All six peers yielded the same results. On the other hand, the therapist obtained an adequate degree of reliability within the subject. This was verified by comparing the previously carried out manipulations carried out to a so-called “gold standard”, in the form of an expert professor with more than 30 years of experience under his belt. This was verified during his training period at the Catholic University of Murcia, by means of an examination with results verified by the professor and obtained through the performance of the Gillet test, the standing forward flexion test, and the seated forward flexion test.

### 2.5. Statistical Analysis

A series of statistical analyses were carried out depending on the nature of the variables to be studied, as well as the distribution that followed each one of them. Firstly, we used the chi-square test in order to check the normal distribution in the groups (amount of participants) and we separated the base date in groups in order to check the normal distribution in sacroiliac joint blockage. The results indicated that the distribution was normal.

The study was centered on analyzing the differences in sacroiliac joint blockage (dependent variable) according to the group (independent variable: MET, thrust or placebo). A chi-square analysis and contingency coefficient were then performed using contingency tables to assess the results, comparing the three groups and their differences. On the other hand, to assess the differences in the 3 data collection moments (pre-intervention, initial intervention, and final intervention), the repetitive measures test was used, previously dividing the database to check this test in each group. The results obtained at the beginning, after the first intervention and after the final manipulation were compared. The IBM SPSS 22.0 statistical package (IBM, Armonk, NY, USA) was used for this purpose. The values of *p* < 0.05 and *p* < 0.001 were regarded as statistically significant as far as the analysis of the results was concerned.

## 3. Results

The differences in the intervention according to the performed technique are shown in Table 3. Values are indicated by relating the two intervention techniques and the differences between them, as well as the differences found between the two intervention techniques and the control group for sacroiliac dysfunction.

An analysis was performed to check the differences according to the sociodemographic variables studied. No statistically significant differences were found for any of the interventions based on the sex, work situation, postural system, injuries or type of sport as categorical variables; nor based on age, body mass or height as quantitative variables.

Table 3 shows that as the number of interventions in the thrust group increased, sacroiliac dysfunction decreased (negative dysfunction). While in the MET group, eight individuals still had a dysfunction (positive dysfunction) and so did 16 individuals in the control group. Statistically significant differences were found comparing the thrust group and the control group (*p* = 0.000) between the pre-intervention and the initial intervention, as well as between the initial and final intervention (*p* = 0.000). In turn, the MET group had statistically significant differences with regard to the control group between the pre-intervention and initial intervention (*p* = 0.000) and between the initial and final intervention (*p* = 0.022). Comparing both techniques, no statistically significant differences were found between the pre-intervention and the initial intervention (*p* = 0.191) but they were confirmed between the initial and the final intervention (*p* = 0.032) in favor of the thrust group. On the other hand, the contingency coefficient showed a value of 0.000 (pre-intervention, the groups had the same quantity of patients with sacroiliac joint blockage), 0.535 (*p* = 0.000, initial intervention), and 0.501 (*p* = 0.000, final intervention). Comparing MET and thrust, the contingency coefficient showed a value of 0.250 (*p* = 0.102) in initial intervention and 0.327 (*p* = 0.028) in final intervention.

The Friedman test indicated that only the thrust technique group obtained statistically affected differences (*p* = 0.000) between the three data captures, thus indicating that for the MET technique, there were no changes in the recent sacroiliac block. The first intervention effected significant changes compared to pre-intervention.

## 4. Discussion

After comparing the measurements made, we observe that the thrust technique offers more significant statistical results than those obtained by the MET technique and the placebo. To our knowledge, this study is the first to compare thrust and MET manipulative techniques in the resolution of sacroiliac dysfunction in a population of athletes. Patel et al. [23] also found that manipulation produced significant improvements compared to MET, even though both techniques were effective in reducing pain and disability in subjects with SIJ dysfunction. However, the researchers did not report whether resolution of the dysfunction or improvement of joint mobility occurred. Thus, most trials of the research analyze the efficacy of these techniques in isolation, in reducing pain and disability in the lower back region due to sacroiliac dysfunction. In our case, we recruited a sample of asymptomatic and physically active individuals [14].

The results indicate the possible existence of a kinetic chain that, by modifying the iliac position, could balance the plantar distribution, thus improving the biomechanics of the SIJ in athletes. Some authors point out that the manipulation of this joint may improve the symmetry of walking and running in individuals with SIJ dysfunctions, since it changes the peripheral control mechanisms such as muscle reflexes and central information processing [15].

Research has evaluated different neurophysiological aspects of SIJ manipulation. Studies that focused on observing the distribution of plantar pressure and the positioning of the center of gravity after a single thrust manipulation in SIJ found significant changes in asymptomatic individuals [15,24,25] Joint manipulation has been investigated in the stimulation of sensory receptors and whether it can affect the central nervous system at the spinal segmental and cortical level. Neurophysiological effects and treatment outcomes in musculoskeletal disorders may depend on the magnitude of forces applied by manual intervention [11,12].

It should be noted that our protocol consisted of a single manipulation in each treatment session. This was aimed at being able to compare the MET, thrust and placebo techniques in an individual way, so that the results would not be affected by performing one intervention right after the previous one. In contrast, Kamali et al. [12] included an additional lumbar manipulation to that of SIJ, obtaining similar results to the isolated thrust treatment—in this case, on the pain and disability of women with SIJ syndrome.

Since our results replicate those in Kamali et al., and dysfunction is corrected in the same manner, we believe it is more beneficial to perform a single thrust technique, as opposed to the two carried out in that previous study.

The results obtained by the intervention group that underwent the MET technique are less successful than those obtained in the group where the thrust technique was performed. The MET technique has been widely used to elongate a shortened, contracted, or spastic muscle or muscle group. It is considered a gentle manual therapy for restricted joint movement and is an active technique in which the patient has control over the corrective force. For many years MET has been advocated as a treatment for lumbopelvic muscle imbalances such as pelvic asymmetry, which may also be present in asymptomatic individuals like our sample. Bindra [26] found that individuals who were treated with this technique showed a greater range of mobility compared to physiotherapy using ultrasound and analgesic electrotherapy. Also, Mathew et al. [18] compared the MET technique to conventional physical therapy mobilization and concluded that MET is more effective in the treatment of SIJ.

Nevertheless, the long-term effect of both techniques as an isolated treatment has not been determined. Carvalho Bardosa et al. [27] used manipulation together with specific exercises for pelvic stability, three sessions per week on non-consecutive days, in patients with painful symptoms. After completion of the intervention, the authors concluded that the combination of both treatments produces a greater decrease in pain.

Thus, as limitations of our study, we can point out that we did not follow up on the efficacy of the techniques in the long term, together with the hampered sample selection, which was affected by accessibility and the reduced sample size. On the other hand, the selection of the sample was for convenience and accessibility. It would have been of great interest to analyze the distribution of body mass between both limbs, both initially and also as a part of the diagnosis. This would have been key in order to see the influence of the blockage on the weight received that lower limbs deal with. Finally, inter-observer reliability might be affected, as there was only one therapist. Nonetheless, one of the strengths of our study refers to the large trial sample, particularly in comparison to most specialized literature. In addition, all participants completed the study, even though they are athletes and their availability is limited.

The musculature surrounding the pelvis is of vital importance to proper biomechanics of running and manipulative therapy can exert its effect on different mechanisms that affect the function of the musculoskeletal system by causing changes in the brain and spinal sensory processing [1].

## 5. Conclusions

In accordance with the results yielded by our Gillet test, the standing forward flexion test, and the seated forward flexion test, the thrust technique is most effective in the treatment of SIJ in middle-distance running athletes. As per these results, a two-time performance of the thrust technique is recommended, with the second manipulation taking place a month after the initial one, the reason being that this yields better results than a single intervention when tackling SIJ dysfunction.

Future lines of research could be devoted to the comparison of thrust and MET manipulative techniques with similar but larger samples, especially of women, in order to compare the differences between sexes, or different samples. Other suggestions for future research relate to the possibility of applying an additional neuromuscular training program in order to enhance activation and rebalancing of athlete’s muscles.

## Figures and Tables

**Figure 1 ijerph-17-04490-f001:**
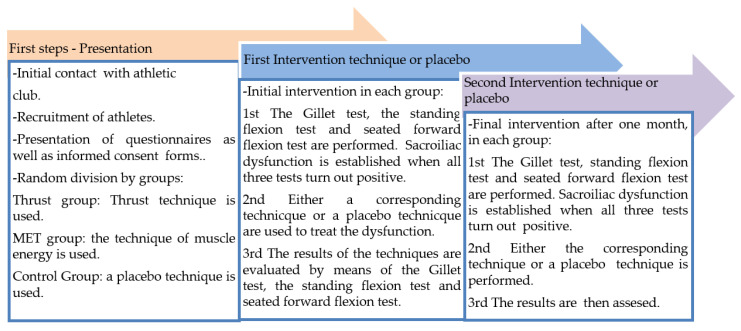
Chronological line of intervention.

**Table 1 ijerph-17-04490-t001:** Descriptive table of continuous variables per group.

Variables	MET				Thrust				Control			
	A	SD	Max	Min	A	SD	Max	Min	A	SD	Max	Min
Age	35.85	8.07	48.00	20.00	31.90	9.65	52.00	19.00	33.85	11.73	62.00	18.00
Body Mass	71.12	13.82	100.00	50.00	71.28	13.25	91.00	53.00	73.35	11.56	93.00	56.00
Height	1.75	0.090	1.97	1.54	1.75	0.09	1.95	1.58	1.75	0.09	1.90	1.60
T.F.	4.85	1.09	6.00	2.00	4.75	1.02	6.00	2.00	4.70	1.13	6.00	3.00
F.P.T.	8.50	7.88	30.00	.00	8.50	8.77	30.00	0.00	7.10	7.59	24.00	0.00

**Note:** A = average; SD = standard deviation; Max = maximum; Min = minimum; T.F. = training frequency in hours per week; F.P.T. = frequency of visits to the physiotherapist on days per year.

**Table 2 ijerph-17-04490-t002:** Descriptive table of categorical variables per group.

Variables		MET		Thrust		Control	
		*n*	%	*n*	%	*n*	%
**Sex**	Male	14	70.0%	14	70.0%	15	75.0%
	Female	6	30.0%	6	30.0%	5	25.0%
**Work situation**	Student	4	20.0%	8	40.0%	6	30.0%
	Worker	14	70.0%	11	55.0%	13	65.0%
	Both of them	2	10.0%	1	5.0%	1	5.0%
**Postural system**	Sitting position	10	50.0%	7	35.0%	9	45.0%
	Standing position	8	40.0%	6	30.0%	4	20.0%
	Both of them	2	10.0%	7	35.0%	7	35.0%
**Dominant leg**	Left	3	15.0%	2	10.0%	1	5.0%
	Right	17	85.0%	18	90.0%	19	95.0%
**Previous injuries**	No	6	30.0%	10	50.0%	11	55.0%
	SI left	7	35.0%	5	25.0%	2	10.0%
	SI right	5	25.0%	2	10.0%	4	20.0%
	Both of them	2	10.0%	3	15.0%	3	15.0%
**Use of insoles**	No	16	80.0%	13	65.0%	19	95.0%
	Yes	4	20.0%	7	35.0%	1	5.0%
**Type of sport**	Running/Athletics	6	30.0%	12	60.0%	10	50.0%
	Running/Athletics + Gym	3	15.0%	5	25.0%	5	25.0%
	Running/Athletics + Others	11	55.0%	3	15.0%	5	25.0%

Note: *n* = number; SI left = sacroiliac left; SI derecha = sacroiliac right.

**Table 3 ijerph-17-04490-t003:** Results of interventions comparing the muscle energy technique (MET) group, the thrust group, and the placebo group.

Time of Intervention		MET		Thrust			Placebo		
		*n*	%	*n*	%	*p*-Value (MET vs. Thrust)	*n*	%	*p*-Value (Placebo vs. MET or Thrust)
Pre-intervention	Positive D.	20	100.0%	20	100.0%		20	100.0%	
	Negative D.	0	0.0%	0	0.0%		0	0.0%	1.000
Initial intervention	Positive D.	10	50.0%	5	25.0%		20	100.0%	
	Negative D.	10	50.0%	15	75.0%	0.191	0	0.0%	0.000 **
Final intervention	Positive D.	8	40.0%	2	10.0%		16	80.0%	
	Negative D.	12	60.0%	18	90.0%	0.032 *	4	20.0%	0.000 **
Friedman *p*-value			0.092		0.000 **			0.155	

Note: *n* = number; *p* = significance; * *p* < 0.05; ** *p* < 0.01; D. = dysfunction.

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
