# Peer review of "Effectiveness of the Muscle Energy Technique versus Osteopathic Manipulation in the Treatment of Sacroiliac Joint Dysfunction in Athletes"

_ijerph, 2020, doi:10.3390/ijerph17124490_

Round 1
Reviewer 1 Report
The authors have substantially improved the manuscript according to the indications made. The reviewer wants to acknowledge that effort. Congratulations.
A few minor considerations:
Line 281: translate the fragment written in Spanish into English.
Line 300: review the ellipsis included and complete the sentence.
Author Response
Thanks you for your appreciations.
We have done the changes (translated the fragment written in Spanish into English and we have complete the sentence in line 300)
Reviewer 2 Report
Abstract
Indicate the total sample and how many participants were in each group. Also indicate how many were men and women.
Indicate dose and duration of each treatment.
Indicate the type of study.
The results of the thrust group compared to those of the energy group were statistically better? Perhaps include concrete results so that in addition to p the reader can understand it better.
Introduction
Paragraph (lines 49-52) move it to line 62 before the muscle energy technique
Change 1st person to 3rd person
2.2 sample:
Indicate the number of females.
Indicate standard deviation of the age of the participants.
Intervention
I am not able to see how long the study lasts and at what times the tests are carried out.
2.5. Statistical Analysis
I think the appropriate test in this case is the Fiedman test. It is a non-parametric 2-factor test (time vs. group). With it you will be able to determine if there were really differences in behavior between groups throughout the intervention to be able to give your results more precisely.
Results
Table 3. Indicate what the 2 p's express? as indicated it seems that it is the same.
Have you tested whether sex is bias? That is, in the 3 groups the number of female patients was similar. Were there even differences between men? I think it is an interesting topic when there are mixed samples.
Discussion Line 281 and 282 review the English
Conclusion: I think it is important to indicate the duration of the treatment
Author Response
Thanks you for your appreciations
We have done the changes:
Abstract
- Indicate the total sample and how many participants were in each group. Also indicate how many were men and women.
- Indicate dose and duration of each treatment.
- Indicate the type of study.
- The results of the thrust group compared to those of the energy group were statistically better? Perhaps include concrete results so that in addition to p the reader can understand it better.
DONE, we have changed the abstract and clarified the results
Introduction
Paragraph (lines 49-52) move it to line 62 before the muscle energy technique
Change 1st person to 3rd person
Has been Changed.
2.2 sample:
Indicate the number of females.
The number of females has been indicated
Indicate standard deviation of the age of the participants.
Standard deviation of the age has been indicated
Intervention
I am not able to see how long the study lasts and at what times the tests are carried out.
The duration of the study and the number of times the test is performed has been added.
2.5. Statistical Analysis
I think the appropriate test in this case is the Fiedman test. It is a non-parametric 2-factor test (time vs. group). With it you will be able to determine if there were really differences in behavior between groups throughout the intervention to be able to give your results more precisely.
You are right, we have done this statistical test and report the results in the table and in the text.
Results
Table 3. Indicate what the 2 p's express? as indicated it seems that it is the same.
Have you tested whether sex is bias? That is, in the 3 groups the number of female patients was similar. Were there even differences between men? I think it is an interesting topic when there are mixed samples.
We have clarified the results. We tested the differences between sex, but, due to the sample of female is small, it can be significative, we have included this in future lines of research at the end of the manuscipt.
Discussion Line 281 and 282 review the English
Done
The duration has been indicated
Round 2
Reviewer 2 Report
The authors have done a great effort to improve the manuscript. However in lines 98-13 the authors should add SD in body mass (no weight) and height. Likewise, the age is diplicate and does not match.
Change weight by body mass thorougth the manuscript.
The authors should indicate in the Statistical Analysis section how and what parameters have included in Friedman test. They do not explain nothing about this.
Author Response
Thanks you so much for your appreciations.
We have analyzed the data base in order to check the right value of body mass, age and height.
We have included in the method, result, and discusion in the manuscript the term of "body mass" instead of "weight"(thanks you so much for this detail).
Finally, the friedman test and the procedure of its use has been included in the 2.5. section.
Thanks you so much
This manuscript is a resubmission of an earlier submission. The following is a list of the peer review reports and author responses from that submission.
Round 1
Reviewer 1 Report
Dear authors,
I have had the opportunity to review his study, entitled “Effectiveness of the muscle energy technique versus 2 osteopathic manipulation in the treatment of 3 sacroiliac joint dysfunction in athletes”. Being an interesting aim of study, there are multiple aspects that should be improved in order to be accepted for publication. The most notable, the lack of reliability statistics to report a strong evidence of the results. In any case, I invite you to read my major revision and consider improving de manuscript.
Introduction
In general terms, the introduction should be improved, giving it a greater explanation of the two techniques used as well as the justification for doing so. In this sense, please check lines 52-53 (at the end of this sentence the “several studies” must be cited). Check lines 53-54 (at the end of this sentence the “studies comparing other techniques… showed conflicting results” must be cited). Please, it would be advisable to review and include the contributions of the following studies, among others…
Peebles R, Jonas CE. Sacroiliac joint dysfunction in the athlete: diagnosis and management. Curr Sports Med Rep. 2017 Sep/Oct;16(5):336-342.
Brolinson PG, Kozar AJ, Cibor G. Sacroiliac joint dysfunction in athletes. Curr Sports Med Rep. 2003 Feb;2(1):47-56.
Nejati P, Safarcherati A, Karimi F. Effectiveness of exercise therapy and manipulation on sacroiliac joint dysfunction: a randomized controlled trial. Pain Physician. 2019 Jan;22(1):53-61.
Al-Subahi M, Alayat M, Alshehri MA, Helal O, Alhasan H, Alalawi A, Takrouni A, Alfaqeh A. The effectiveness of physiotherapy interventions for sacroiliac joint dysfunction: a systematic review. J Phys Ther Sci. 2017 Sep;29(9):1689-1694.
Sarkar M, Goyal M, Samuel AJ. Comparing the effectiveness of the muscle energy technique and kinesiotaping in mechanical sacroiliac joint dysfunction: a non-blinded, two-group, pretest-posttest randomized clinical trial protocol. Asian Spine J. 2020 Jan 30. [Epub ahead of print]
Material and Methods
This section is messy, confused and too extensive. It should be defragmented in subtitles or subsections, being recommended to include them in the following order:
Design research (include all concerning to research design (randomized strategy…) and consider citing the independent and/or dependent variables analyzed with its related unit of measure). What is the unit of measurement for each tests?
Sample (include sample characteristics, inclusion/exclusion criteria, ethical considerations)
Material and procedures (include test description or, in this case, both techniques used, protocols, material… and, ideally, draw a figure that clarifies (as a timeline) the chronological temporality of the procedures performed.
Statistical analysis
Additionally, in this section it is necessary to clarify:
- The original studies that demonstrate the validation of each of the 3 tests used. Lines 75-77: the justification for the use of these tests is not enough with the references provided.
- In the sample section, is necessary to describe by groups of analysis and sex: (chronological age, weight, and stature).
- Where are the mechanisms to control the reliability of the tests? Intra and Inter-rater reliability statistics (ICC, CV, etc.) must be reported. In this point, this is a critical gap in the research design (single manipulation in each treatment session… please justify it, beyond what is indicated on lines 208-211 in the discussion section…). This is an obvious limitation of the study.
Results
Consider relocating, totally or partially, text between lines 147-151 (as well as Table 1), in the sample description subsection (Material and Methods section).
The variables show in Table 2: “work situation: student, worker, both of them”; “postural system… idem), and others, is the first time that they appear on the manuscript. Its relevance in the methodological structure of this study must be explained in the corresponding previously section (Material and Methods).
Discussion
Line 197-200… some authors… ¿? Please, cite and reference.
Line 204-207… joint manipulation has been investigated… ¿? Please, cite and reference.
Conclusions
The authors consider that they can be so emphatic/rotund when it comes to affirming that the thrust technique is most effective in the treatment of SIJ in 238 middle-distance running athletes, without any reliability test?? Please reconsider and rewrite the limitations and conclusions sections.
Author Response
Thanks for your contributions, we have tried to improve the manuscript attending to all your suggestions.
Introduction
In general terms, the introduction should be improved, giving it a greater explanation of the two techniques used as well as the justification for doing so. In this sense, please check lines 52-53 (at the end of this sentence the “several studies” must be cited). Check lines 53-54 (at the end of this sentence the “studies comparing other techniques… showed conflicting results” must be cited). Please, it would be advisable to review and include the contributions of the following studies, among others…
Peebles R, Jonas CE. Sacroiliac joint dysfunction in the athlete: diagnosis and management. Curr Sports Med Rep. 2017 Sep/Oct;16(5):336-342.
Brolinson PG, Kozar AJ, Cibor G. Sacroiliac joint dysfunction in athletes. Curr Sports Med Rep. 2003 Feb;2(1):47-56.
Nejati P, Safarcherati A, Karimi F. Effectiveness of exercise therapy and manipulation on sacroiliac joint dysfunction: a randomized controlled trial. Pain Physician. 2019 Jan;22(1):53-61.
Al-Subahi M, Alayat M, Alshehri MA, Helal O, Alhasan H, Alalawi A, Takrouni A, Alfaqeh A. The effectiveness of physiotherapy interventions for sacroiliac joint dysfunction: a systematic review. J Phys Ther Sci. 2017 Sep;29(9):1689-1694.
Sarkar M, Goyal M, Samuel AJ. Comparing the effectiveness of the muscle energy technique and kinesiotaping in mechanical sacroiliac joint dysfunction: a non-blinded, two-group, pretest-posttest randomized clinical trial protocol. Asian Spine J. 2020 Jan 30. [Epub ahead of print]
Response 1: First of all, we would like to stress how really appreciate we are of every one of the considerations you have suggested in order to improve the document. The introduction has been changed, following the reviewer's recommendations, the explanation of the techniques and the justification for them have also been improved. We’ve added new bibliographic justification. References and bibliographic references have been added to lines 52-53 as well as lines 53-54. Similarly, we have used bibliography provided by the reviewer as well as new one on our part.
Material and Methods
This section is messy, confused and too extensive. It should be defragmented in subtitles or subsections, being recommended to include them in the following order:
Response 2: Following the reviewer's recommendations, the structure of the "Materials and methods" section has been modified. Adding subsections as: "Design Research", "Sample", "Material and procedures", "Diagnosis", "Intervention", "Fidelity of Implementation" and "Statistical Analysis". Changes have also been made to the contents and structure of the subparagraphs in order to facilitate their reading and understanding.
Design research (include all concerning to research design (randomized strategy…) and consider citing the independent and/or dependent variables analyzed with its related unit of measure). What is the unit of measurement for each tests?
Response 3: Done. The subsection and all the information recommended by the reviewer have been added. The variables, as well as the units of measurement of each variable, are also cited.
Sample (include sample characteristics, inclusion/exclusion criteria, ethical considerations)
Response 4: Done. The subsection and all the information recommended by the reviewer have been added. The information on the characteristics of the sample, the ethical considerations and the inclusion/exclusion criteria are also added.
Material and procedures (include test description or, in this case, both techniques used, protocols, material… and, ideally, draw a figure that clarifies (as a timeline) the chronological temporality of the procedures performed.
Response 5: Done. The subsection and all the information recommended by the reviewer have been added. In addition, following the reviewer's recommendations, we added a figure as a timeline.
Statistical analysis
Response 6: Done. The subsection and all the information recommended by the reviewer have been added.
Additionally, in this section it is necessary to clarify:
- The original studies that demonstrate the validation of each of the 3 tests used. Lines 75-77: the justification for the use of these tests is not enough with the references provided.
- In the sample section, is necessary to describe by groups of analysis and sex: (chronological age, weight, and stature).
- Where are the mechanisms to control the reliability of the tests? Intra and Inter-rater reliability statistics (ICC, CV, etc.) must be reported. In this point, this is a critical gap in the research design (single manipulation in each treatment session… please justify it, beyond what is indicated on lines 208-211 in the discussion section…). This is an obvious limitation of the study.
Response 9: Done. We have explained in the section "Fidelity of Implementation" how the reliability and training of the research therapist who performed the techniques and tests were verified. In addition, the discussion has justified why a single technique is performed per group and intervention.
Results
Consider relocating, totally or partially, text between lines 147-151 (as well as Table 1), in the sample description subsection (Material and Methods section).
Response 10: Done.
The variables show in Table 2: “work situation: student, worker, both of them”; “postural system… idem), and others, is the first time that they appear on the manuscript. Its relevance in the methodological structure of this study must be explained in the corresponding previously section (Material and Methods).
Response 11: Done. In accordance with the reviewer's recommendations, explanations of variables have been added in the subsection "Sample" under the heading "Materials and Methods".
Discussion
Line 197-200… some authors… ¿? Please, cite and reference.
Line 204-207… joint manipulation has been investigated… ¿? Please, cite and reference.
Response 13: Done.
Conclusions
The authors consider that they can be so emphatic/rotund when it comes to affirming that the thrust technique is most effective in the treatment of SIJ in 238 middle-distance running athletes, without any reliability test?? Please reconsider and rewrite the limitations and conclusions sections.
Response 14: Done. According to the reviewer, we have revised and rewritten the "conclusions" section. In it, we have taken a different approach to conclusion and added the limitations.

Reviewer 2 Report
Abstract
Include some background in this section previous the objective
Delete (1), (2)….
Include 0.032 in p
Change p = .000 to p<0.001
Introduction
Lines 34, 37, 40, 43, 53,54. Include sone references to justify these affirmations.
Line 59. Change 1st person by 3rd person in all manuscript
Line 62. Use the past. Change is by was
Materials and methods
This section should be divided into some subsections. It is very long and heavy to read.
Which was the study methodology? Double blind placebo controlled,…?
How long were the treatments?
Conclusions
Explain better this conclusion. Explain the dose, time,… of the different technics…
Author Response
Thanks for your contributions, we have tried to improve the manuscript attending to all your suggestions. Abstract
Include some background in this section previous the objective
Response 1: First of all, we really appreciate every one of the considerations you have done to improve the document. Furthermore following the reviewer's recommendations, we have added more background to the abstract section.
Delete (1), (2)….
Response 2: Done
Include 0.032 in p
Response 3: Done
Change p = .000 to p<0.001
Response 4: Done
Introduction
Lines 34, 37, 40, 43, 53,54. Include sone references to justify these affirmations.
Response 5: Done
Line 59. Change 1st person by 3rd person in all manuscript
Response 6: Done
Line 62. Use the past. Change is by was
Response 7: Done
Materials and methods
This section should be divided into some subsections. It is very long and heavy to read.
Response 8: Following the reviewer's recommendations, the structure of the "Materials and methods" section has been modified. Adding subsections as: "Design Research", "Sample", "Material and procedures", "Diagnosis", "Intervention", "Fidelity of Implementation" and "Statistical Analysis". Changes have also been made to the contents and structure of the subparagraphs to facilitate their reading and understanding.
Which was the study methodology? Double blind placebo controlled,…?
Response 9: In the subsection "Design Research" we have tried to respond to this by defining the study as "a quasiexperimental study, with pre- and post-intervention measurements..."
How long were the treatments?
Response 10: An initial intervention is made and another one is made per month so as to observe if the techniques are more effective, and in order to assess which of them is more effective. This information has been added in the "abstract" and "conclusions" sections and has also been added in the "intervention" section of the "materials and methods" section.Conclusions
Conclusions
Explain better this conclusion. Explain the dose, time,… of the different technics…
Response 11: Done. According to the reviewer, we have corrected the "conclusions" section. We have taken a different approach to the conclusion, where we have added the recommendations for the implementation of the thrust technique to make it more effective. We have also added the limitations of the study.
We have carried out with a specialized translator a new revision of the English language.

Round 2
Reviewer 1 Report
I have attached a file with my corrections from this second review.

Reviewer 2 Report
The authors have made a great effort to improve the article.